# Interpretable Matching of Optical-SAR Image via Dynamically Conditioned Diffusion Models

Shuiping Gou
shpgou@mail.xidian.edu.cn
Xidian University
Xi'an, China

Xin Wang
23171214441@stu.xidian.edu.cn
Xidian University
Xi'an, China

Xinlin Wang*
wangxinlin@xidian.edu.cn
Xidian University
Xi'an, China

Yunzhi Chen
2016010026@hzvtc.edu.cn
Hangzhou Vocational and Technical College
Hangzhou, China

## Abstract

Driven by the complementary information fusion of optical and synthetic aperture radar (SAR) images, the optical-SAR image matching has drawn much attention. However, the significant radiometric differences between them imposes great challenges on accurate matching. Most existing approaches convert SAR and optical images into a shared feature space to perform the matching, but these methods often fail to achieve the robust matching since the feature spaces are unknown and uninterpretable. Motivated by the interpretable latent space of diffusion models, this paper formulates an optical-SAR image translation and matching framework via a dynamically conditioned diffusion model (DCDM) to achieve the interpretable and robust optical-SAR cross-modal image matching. Specifically, in the denoising process, to filter out outlier matching regions, a gated dynamic sparse cross-attention module is proposed to facilitate efficient and effective long-range interactions of multi-grained features between the cross-modal data. In addition, a spatial position consistency constraint is designed to promote the cross-attention features to perceive the spatial corresponding relation in different modalities, improving the matching precision. Experimental results demonstrate that the proposed method outperforms state-of-the-art methods in terms of both the matching accuracy and the interpretability.

## CCS Concepts

• **Computing methodologies** → **Artificial intelligence**; **Computer vision**; **Computer vision problems**; **Matching**.

## Keywords

Image matching, Synthetic aperture radar images, Diffusion probabilistic model

---

*Corresponding author.

**ACM Reference Format:**
Shuiping Gou, Xin Wang, Xinlin Wang, and Yunzhi Chen. 2024. Interpretable Matching of Optical-SAR Image via Dynamically Conditioned Diffusion Models. In *Proceedings of the 32nd ACM International Conference on Multimedia (MM '24), October 28-November 1, 2024, Melbourne, VIC, Australia.* ACM, New York, NY, USA, 10 pages. https://doi.org/10.1145/3664647.3681700

## 1 Introduction

With the rapid development of remote sensing technology, different image sensors are constantly emerging and provide rich image data for the earth observation. Owing to the complementary property between different modality images, the multi-source information integration is widely explored. Especially, the synthetic aperture radar (SAR) and optical sensing images have been increasingly explored and applied to mapping [45], object detection [36], and etc. Therefore, effectively integrating and exploiting optical and SAR images is the focus, in which the optical-SAR image matching is the core issues. However, due to their different imaging mechanisms, there exist remarkable geometric differences and nonlinear radiometric variations between optical and SAR images. As a result, the optical and SAR image matching still remains challenging.

To solve the task, researchers devote themselves to the challenge and propose various algorithms. In the early stages, traditional image matching methods are widely used, including region-based and feature-based methods. Region-based methods use similarity metrics, such as normalized mutual information (NMI) [2] and normalized cross-correlation (NCC) [20], to find the correspondence between the template and the reference images. The kind of approaches only leverage the global pixel intensity information within a window to calculate the similarity of the corresponding region, which is sensitive to the image intensity differences and the noise. To this end, feature-based image matching methods are developed, which aims to extract keypoints of each image and find matching points. The representative algorithm is the scale-invariant feature transform (SIFT) [23], which detect keypoints in different scales by utilizing its description in terms of the scale, the gradient magnitude, and direction. Based on the SIFT algorithm, some variants, such as BFSIFT [38], AAGSIFT [37], and RIFT [21], have been derived. In addition, to overcome the nonlinear radiometric differences in SAR and optical images, structural similarity-based descriptors have been proposed, such as histogram of oriented phase coherence (HOPC) [43], and channel feature of oriented gradients (CFOG) [42]. However, the potential of hand-designed features for

improving the matching performance is very limited considering the remarkable differences between SAR and optical images.

Thanks to the considerable achievements made by the convolutional neural networks (CNNs) in computation vision tasks, and the launch of the publicly available remote sensing data [31, 34, 41], deep-learning-based cross-modal image matching approaches have constantly emerged[16, 24–26, 44]. Existing deep-learning-based optical-SAR image matching methods includes two kinds: the Siamese network-based feature mapping, and the generative adversarial network(GAN)-based cross-modality translation. The Siamese network-based methods [24–26, 44] exploit the effective feature extraction ability of CNN to map the multi-modal images into the common feature spaces, and perform similarity metrics on the mapped features spaces. However, the mapping feature spaces are not visual, and lack of interpretability. Furthermore, these methods fail to extract useful features for the matching when there are more textureless regions in the images. In contrast, the GAN-based approaches [13, 28] attempt to translate the SAR or optical image from one modality to the other by the adversarial learning between generators and discriminators. But, GAN trains the model through the mutual game between the generator and discriminator, which easily causes the model to fall into a local minimum, generating unstable translations on the cross-modal images with large differences. Moreover, GAN-based matching methods are not end-to-end.

Recently, the diffusion model [17] has received considerable attention on generative models, which has a diffusion process to gradually add Gaussian noise to the data and a denoising process to learn to remove it. Driven by its stable training manner and high-quality generation results, the diffusion models have been explored to the optical-SAR image translation. Bai et al. [5] utilizes a conditional diffusion model to efficiently translate SAR images into optical images, and Shi et al. [32] proposes self-attention and long skip connections in denoising networks to enhance feature extraction, which demonstrates the potential of diffusion models in translating SAR images. However, existing diffusion model-based optical-SAR translation methods only focus on generating cross-modal images for enhancing human visual perception, and have not yet explored the downstream cross-modal image matching task. In fact, using the generated image for matching requires elaborate design of matching methods, since there still exist different attributes between the generated remote sensing images and the real images, which deteriorates the matching performance and the speed.

To address the above problem, this paper formulates the optical-SAR cross-modal image matching as a dynamically conditioned diffusion model (DCDM), which aims to learn the posterior distribution of regions with dense correspondences. Specifically, the optical template and SAR search image pairs are taken as conditions to respectively provide the content for the better generation and the texture details for the accurate matching. Moreover, a gated dynamic sparse cross-attention (GDSC) module is designed to dynamically inject reliable conditional information into the generative network and accelerate denoising process. On this basis, to enhance the perception of matching positions, we introduce the spatial position consistency constraint. In the matching, to reduce the computational effort, the latent features of the generated SAR are directly matched with features of the search SAR, instead of

decoding them into images and then matching them. The contributions of this paper are summarized as follows:

- We propose an end-to-end cross-modal image matching framework, dynamically conditioned diffusion model (DCDM). It not only translates cross-modal images, but also completes the pixel-level matching in the latent space.
- A gated dynamic sparse cross-attention module is present to perform the controlled and efficient cross-interaction between the template and the search, aiming to filter out the outlier matching regions while improving the computational efficiency.
- A spatial position consistency constraint is designed to enhance the detail perception of the cross-attention to generate more accurate cross-modal features for matching. Experimental results on two matching datasets quantitatively and qualitatively demonstrates the effectiveness and the interpretability of the proposed approach.

## 2 RELATED WORKS

### 2.1 Learning-based multimodal remote sensing image matching

In learning-based multimodal image matching models often leverage intensive interactions between two modalities to capture effective matching features. The studies [7, 14] apply the cross attention to perform long-range interactions of cross-modal features, thus capturing features suitable for matching. Other methods [9, 26] slide the template features on the reference features pixel by pixel to calculate the similarity heatmap, which is computationally intensive, especially for large images. Fang et al. [12] leverages U-Net [30] to extract high-resolution features, and use the Fast Fourier Transform (FFT) to implement the NCC similarity metric in the frequency domain to accelerate, which has been widely used. Mu et al. [26] proposes a two-stage feature extraction network to achieve precise localization from coarse to fine, which designs a suppression network and a triple loss to suppress false matching position. Zhang et al. [44] presents to fuse low-level fine-grained localization features with high-level semantic features to enhance feature discrimination. Michele et al. [15] extracts features from full-size and half-size images, and then fuses these features to construct pixel-level features for matching.

Despite the advances in matching accuracy, these methods still face challenges of the uninterpretable feature spaces, the indiscriminate matching for textureless regions, and the large amounts of computation requirements. In contrast, we propose the gated dynamical sparse attention under the latent diffusion paradigm to efficiently extract cross-modal features with consistent representation, achieving interpretable and robust matching.

### 2.2 Denoising diffusion model

The emergence of the denoising diffusion probabilistic model (DDPM) [17] has led to the widespread use of diffusion models in computer vision [4], natural language processing [3], interdisciplinary applications [1], and audio processing [18], which outperforms the current GAN-based generative models in image synthesis [27]. DDPM is a parametric Markov chain that incrementally adds noise to the

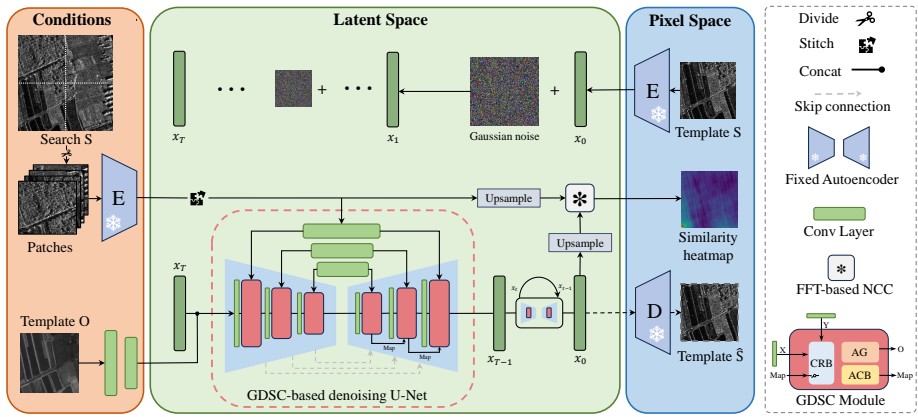

**Figure 1: The pipeline of the proposed dynamically conditioned diffusion model, which aims to generate and match SAR template in the latent space conditioned on the optical template and the search SAR images.**

data during forward diffusion until the original signal is completely corrupted, and then reconstructs the signal during reverse diffusion. The denoising diffusion implicit model (DDIM) [33] is evolved from DDPM, which introduces non-Markov chain diffusion process. This innovation reduces the number of steps required in the inference process. Furthermore, to train denoising diffusion models on limited computational resources while maintaining their quality and flexibility, the latent diffusion model (LDM) [29] applies DDPM to the latent space extracted by the powerful pre-trained autoencoders. Compared to other diffusion models, LDM significantly reduces computational requirements, and achieves efficient cross-modal generation. Therefore, our approach adopts LDM as the base framework to develop the optical-SAR matching algorithm.

## 2.3 Sparse Attention

Over the past few years, the transformers have been exploded in the computer vision community[6, 10]. Contrary to the convolution operation that extract local features, transformers exploit self-attention mechanisms to capture long-distance dependencie, and have global receptive fields[35]. However, such a property comes at the cost of having a high computational complexity and a large memory footprint. To mitigate this problem, the sparse attention [8] has been proposed, in which each query focuses on only a small number of key-value pairs instead of all key-value pairs. Several hand-crafted sparse patterns have been proposed, such as limiting attention in localized windows [22], expanding the windows [39], and etc. Recently, a novel dynamic sparse attention, Biformer, [46] has been proposed, whose two-layer routing architecture performs the dynamic computational allocation through information awareness, effectively reducing the computational complexity.

All of the above approaches focus on designing sparse self-attention by reducing the number of key-value tokens. In fact, for the cross-attention in remote sensing image matching task, filtering queries related to the template is also important, since invalid queries bring interferences. Hence, this paper presents a gated dynamic sparse cross-attention by dynamically selecting both effective queries and keys for efficient computation and matching.

## 3 THE PROPOSED METHOD

To achieve the cross-modal optical-SAR image matching, we formulate a dynamically conditioned diffusion model to translate the optical templates into SAR templates, and perform the matching. The pipeline of the proposed method is illustrated in Figure 1. Firstly, we continuously add Gaussion noise to the ground-truth SAR template in each diffusion step, and then generate the SAR template via training a U-Net-based denoising network. To generate more realistic SAR images, the corresponding optical template is adopted as a condition to provide the scene content. Simultaneously, a task-oriented condition, the search SAR image, is introduced to provide the texture details for the accurate matching. Afterwards, apply the gated dynamic sparse cross-attention module and the spatial position consistency constraint to achieve the effective and efficient cross-modal feature interaction and aggregation. Finally, FFT-based NCC [12] is adopted to perform the matching between the generated SAR template and the search SAR in the latent space.

## 3.1 SAROPT-conditioned Latent Diffusion

The optical-SAR cross-modal image matching task aims to find the corresponding spatial position of the optical image in the SAR image. To reduce the influence of modality differences, this paper proposes a dynamically conditioned diffusion model (DCDM) to formulate the generative process to translate optical images. However, using noise only to generate SAR image is intractable, since remote sensing images contains rich targets. Fortunately, optical images contain object information, which can provide the content information. Thus, the optical template $T_O$ is treated as the condition to guide the generation of the real SAR template. In addition, the texture details is important for the matching. Considering that the generated SAR template is matched with the search SAR, we further take the search SAR image as another condition to supplement the texture details.

However, denoising in pixel space is time-consuming and resource-shortcoming. Inspired by the LDM [29], we utilize an auto-encoder [11] to learn a latent space of the perceptual compression of SAR images, and perform diffusion process in latent space to reduce

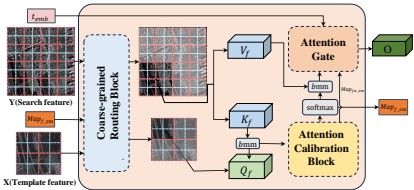

**Figure 2: The gated dynamic sparse cross-attention module.**

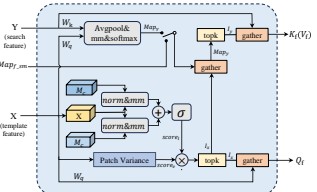

**Figure 3: The coarse-grained routing block.**

the computational complexity. Specifically, the encoder is respectively applied to the real SAR template image $T_S$ and the search SAR image patches $S_S$ to obtain their emdeddings $x_0$, and $c_{sar}$. Correspondingly, the optical template $T_O$ is encoded to $c_{opt}$ by two convolutional layers. Notable, to speed up the perceptual compression, the search SAR images are splitted into smaller patches $S_S$ to input the encoder, and stitched after the encoding. In the forward diffusion process, continuously add Gaussian noise into $x_0$ in each diffusion step to obtain $x_t$. In the reverse diffusion, denoise $x_t$ given conditions $c_{opt}$ and $c_{sar}$, which is expressed as:

$$p_\theta \left( x_{t-1} \mid x_t, c_{opt}, c_{sar} \right) = N \left( x_{t-1}; \mu_\theta \left( x_t, t, c_{opt}, c_{sar} \right), \sigma_t^2 \mathbf{I} \right) \quad (1)$$

The SAROPT-conditioned latent diffusion model not only increases the denoising speed, but also make the matching to be end-to-end, instead of constructing the SAR template and then extracting features for the matching.

### 3.2 Gated Dynamic Sparse Cross-Attention

Despite denoising in the latent space, it is still requires a large computations in conditional features interaction. Particularly, the search image is relatively larger. In addition, conditional remote sensing images contains a large number of textureless regions, resulting in tedious similarity calculation, and even a negative effect. Therefore, we present a gated dynamic sparse cross-attention module to dynamically select effective regions to perform the cross-modal conditional interaction, shown in Figure 2. It firstly exploits the coarse-grained routing block (CRB) to efficiently calculate the sparse cross-attention map, and then utilizes the attention calibration block (ACB) to smooth the cross-attention map, and finally leverages an attention gate (AG) to control the preference between the conditional information interaction and the denoising generation. The details are elaborated on the following.

***Coarse-Grained Routing Block.*** Recently, Biformer [46] presents the sparse self-attention to save both computation and memory. Benefiting from the idea, we downsample the feature map in the U-Net encoder to obtain cross attention on coarse-grained feature map, and then filter out redundant similar feature regions. Afterwards, the fine-grained cross attention is performed on the remaining feature regions, thus reducing the computational overhead.

As shown in Figure 3, the template feature maps $X$, and the search image feature maps $Y$ are respectively divided into $s$ and $S$ feature patches as $X \in \mathbb{R}^{s \times P \times C}$ and $Y \in \mathbb{R}^{S \times P \times C}$. Afterwards, the query $Q$, key $K$, and value $V$, are calculated through the linear projection $W_q$, $W_k$, $W_v$:

$$Q = XW_q, \quad K = YW_k, \quad V = YW_k \quad (2)$$

Then, find patches in the $K$ that should be attended for each given patch in the $Q$. Specifically, we obtain the coarse-grained features of $Q$ and $K$ in the encoder, denoted as $Q_c \in \mathbb{R}^{s \times C}$ and $K_c \in \mathbb{R}^{S \times C}$. Herein, the average of each feature patch is adopted to integrate its information. Thus, the importance of feature patches of projected features $K$ of $Y$ can be represented by the coarse-grained cross-attention map $Map_Y$, formulated as:

$$Map_Y = \frac{bmm \left( Q_c, K_c^T \right)}{\sqrt{C}} \quad (3)$$

Notably, in the decoder stage, we directly input the cross-attention map of the previous layer into the next layers, instead of recalculating the coarse-grained cross-attention map of next layer high-resolution features. It not only avoids duplicate calculations, but also exploits the cross-attention map obtained by rich semantics.

Besides, we calculate the importance of each feature patch in the template features $X$. Different from the calculation of $Y$, we define a score to measure the important feature patches of $X$. It is known that patches with rich matching information tend to have higher variance values. Hence, the variance of each patch is adopted as a metric of information importance. But, some outlier feature patches inevitably obtain high variances, which causes negative effects. To avoid selecting patches with high variances but side effects, we further exploit two learnable weights, $M_c$ and $M_s$, to dynamically interact with each feature patch of $X$ in the spatial and channel dimensions, respectively. Thus, a learnable importance score is obtained. Finally, use the learnable score to refine the variance score. The process is expressed as follows:

$$score_v = \frac{1}{C \cdot P} \sum_{j=1}^{C} \sum_{i=1}^{P} \left( X_{i,j} - \frac{1}{P} \sum_{i}^{P} X_{i,j} \right)^2$$

$$score_l = \text{Sigmoid} \left( \text{Mean} \left( XM_c \right) + \text{Mean} \left( X^T M_s \right) \right) \quad (4)$$

$$score_X = score_v \times score_l$$

where $score_v$ is the variance measured importance, $score_l$ is the learned importance, and $score_X \in \mathbb{R}^{1 \times s}$ is the refined importance metric. $X \in \mathbb{R}^{s \times P \times C}$ represents template features. $M_c \in \mathbb{R}^{C \times M}$ and $M_s \in \mathbb{R}^{P \times M}$ denote learnable weights of linear layers (1D-Conv), and $M$ is the specified dimension. In experiments, we set $M = C$.

Based on the feature patch importance $Map_Y \in \mathbb{R}^{s \times S}$ and $score_X \in \mathbb{R}^{1 \times s}$ of $Y$ and $X$, the feature patches are selected. In experiments, we set a hyperparameter $\gamma$ (0~1) to control the number of selected tokens. Assuming that the number of feature patches of a feature map is $N$, the top $N \times \gamma$ tokens, i.e., $N_\gamma$ tokens are selected. Moreover, $\gamma$ is in inverse proportion to the feature resolution to ensure

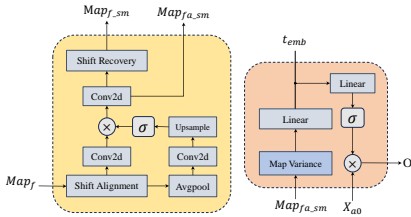

**Figure 4: The attention calibration block and the attention gate.**

smaller computation and progressive refinement in the decoder. The token selection process is formulated as:

$$I_x = \text{topkIndex}(score_X)$$
$$Map_y = \text{gather}(Map_Y, I_x) \quad (5)$$
$$I_y = \text{topkIndex}(Map_y)$$

where 'topkIndex' is used to obtain the indexes of top $K$ values in a feature map. 'gather' is used to select feature patches based on the index. $I_x \in \mathbb{R}^{s_\gamma}$ is the indexes of the top $s_\gamma = s \times \gamma$ feature patches in $X$. $Map_y$ represents the selected row tokens of $Map_Y$. $I_y \in \mathbb{R}^{s_\gamma \times S_\gamma}$ is indexes of $S_\gamma$ patches in $Y$ that should be attended for each patch in the sparsed $X$. Based on the coarse-grained index matrix $I_x$ and $I_y$, the fine-grained projection features of the query, key, and value are obtained:

$$Q_f = \text{gather}(Q, I_x)$$
$$K_f = \text{gather}(K, I_y) \quad (6)$$
$$V_f = \text{gather}(V, I_y)$$

Then, the sparse fine-grained cross-attention map $Map_f \in \mathbb{R}^{s_\gamma P \times S_\gamma P}$ ($S_\gamma P = S_\gamma \times P$) is calculated using the $Q_f \in \mathbb{R}^{s_\gamma \times P \times C}$, $K_f \in \mathbb{R}^{s_\gamma \times S_\gamma P \times C}$, and $V_f \in \mathbb{R}^{s_\gamma \times S_\gamma P \times C}$:

$$Map_f = \frac{bmm\left(Q_f, K_f{}^T\right)}{\sqrt{C}} \quad (7)$$

where $bmm$ is the batch matrix multiplication, the $T$ denotes the transpose operation.

**Attention Calibration Block.** All the selected coarse-grained feature patches have similar semantics, but the corresponding multiple fine-grained features contain different information, which lead to the fact that the fine-grained features in the same coarse-grained feature patch focus on inconsistent positions after cross-attention. To overcome the problem, we use a calibration module to locally smooth the cross-attention map, as shown in Figure 4.

Due to the fact that the cross-attention map represents the matching position of template features $Q_f$ to the search features $K_f$, the distribution of highlighted values in the cross-attention map is diagonal sparse and discrete, shown in Figure 5. Therefore, each row of the cross-attention map $Map_f$ is firstly circularly shifted to achieve spatial alignment attention map $Map_f$ according to $I_x$ and $I_y$. Then, three convolution operations $K_1 \in \mathbb{R}^{1 \times C}$, $K_2 \in \mathbb{R}^{1 \times C}$, and $K_3 \in \mathbb{R}^{C \times 1}$ are performed on the spatially aligned attention map

for smoothing, expressed as:

$$Map_{fa} = \text{Align}\left(Map_f\right)$$
$$A_w = \text{Sigmoid}\left(\text{Up}(\text{AvgPool}(Map_{fa}) * K_1)\right) \quad (8)$$
$$Map_{fa\_sm} = \left(\left(Map_{fa} * K_2\right) \cdot A_w\right) * K_3$$

where $\text{AvgPool}(\cdot)$ is average pooling with pooling size 2×2, and its step size is 2. $\text{UP}(\cdot)$ is a bilinear interpolation.

After obtaining the smoother attention map, the cross-attention map is reset to the origin spatial position by cyclic shifting to calculate the attention features, represented as:

$$Map_{f\_sm} = \text{InvAlign}\left(Map_{fa\_sm}\right)$$
$$X_a = \text{bmm}\left(\text{Softmax}(Map_{f\_sm}), V_f\right) \quad (9)$$
$$X_{a0} = \text{Padding}(X_a, 0)$$

where $X_{a0}$ denotes the features padded the unselected feature regions with 0. It has the same size as the original feature $X$. As a note, the calibration module is lightweight, since it performed on 2-D cross-attention map.

**Attention Gate.** The remote sensing images often contain plain areas, which is textureless. Hence, using the kind of images as conditions is ineffective and even cause a negative effect on the denoising generation. In the situation, the denoising network should focus on using the input itself to autonomously generate, instead of relying on the information in conditions. Considering that, we design an attention gate to control the inflow of conditional information, as shown in Figure 4.

In detail, the smaller the variance of the spatially aligned cross-attention map, the better the attention. Hence, the variance is calculated to measure the attention effect. Moreover, conditional features at different steps have different perceptual abilities, the time step $t$ is also introduced to control the usage of conditions in $t^{th}$ step. Given the spatially aligned and smoothed cross-attention map $Map_{fa\_sm} \in \mathbb{R}^{I \times J}$ ($I = s_\gamma P$, $J = S_\gamma P$), the step time embedding $t_{emb} \in \mathbb{R}^{C/2}$, the linear encoding layers $L_1 \in \mathbb{R}^{1 \times C/2}$ and $L_2 \in \mathbb{R}^{C \times C}$, the attention gate is expressed as:

$$var = \frac{1}{I \times J} \sum_{j=1}^{J} \sum_{i=1}^{I} \left(Map_{fa\_sm_{i,j}} - \frac{1}{I} \sum_{i}^{I} Map_{fa\_sm_{i,j}}\right)^2 \quad (10)$$
$$O = X_{a0} \cdot \text{Sigmoid}\left(L_2\left(\text{concat}\left[t_{emb}, L_1\left(var\right)\right]\right)\right)$$

### 3.3 Spatial position consistency constraint

There exists great scattering characteristic differences in cross-modal images, the random matching position, and multiple similar matching, which leads to incorrect attention. Therefore, the paper further design a spatial position consistency constraint to constrain the correspondence relationships in the cross-attention map. We use the cross-entropy loss to constrain the predicted heatmap $\hat{p}$ and ground-truth attentional map $p$, express as:

$$L_{spcc} = -\sum_i p_i \cdot \log\left(\hat{p}_i\right) + (1 - p_i) \cdot \log\left(1 - \hat{p}_i\right) \quad (11)$$

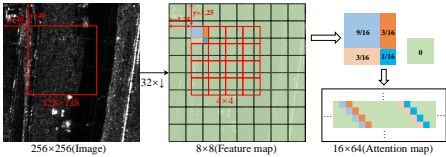

**Figure 5: The generation of ground-truth cross-attention maps.**

The ground-truth attention map $p$ is generated as shown in Figure 5. Assuming that the template image (the red box) (128×128) is matched on the position of (x=40, y=40) in the search image (256×256), the ground-truth matching position in the 32× downsampling feature map is a decimal. In general, we regard the attention value of the matching region is 1, while the non-matching region's is 0. Hence, we divide 1 into different weights according to the ratio of each pixel of the template (4×4) to each pixel of the search (8×8), resulting in 16×64 ground truth matches values to constrain the attention map.

### 3.4 Loss functions

The optimization of the proposed dynamically conditioned diffusion model involves two parts: the denoising network, and the spatial position consistency constraint. To optimize the denoising network, the $L2$ loss and the SSIM[40] loss are adopted, expressed as:

$$L_{den} = \|f_\theta (x_t, t) - x_0\|^2 + \text{SSIM} (f_\theta (x_t, t), x_0) \qquad (12)$$

In summary, the total loss $L$ is the sum of the denoising loss $L_{den}$ and the spatial position consistency constraint loss $L_{spcc}$ (refer to Equation (11) ):

$$L = L_{den} + L_{spcc} \qquad (13)$$

## 4 EXPERIMENTS AND ANALYSES

### 4.1 Datasets

**Sentinel-1 and Sentinel-2 (SEN1-2) datasets:** The SEN1-2 dataset [31] contains a total of 282,384 aligned SAR and optical image pairs with the size of 256×256. The SAR images are acquired from the dual-polarized SAR data of Sentinel-1. The optical images are acquired from the multispectral images of Sentinel-2. The fourth, third and second bands are used to generate RGB images. The dataset is collected from four seasons. The images have a spatial resolution of 10m. In experiments, 4088 image pairs are randomly selected from the spring season data for evaluation. The training, testing and validation sets are splitted by a ratio of 7:2:1. The SAR images with the size of 256×256 are treated as the search images, and the optical images cropped 128×128 as the template.
**OSdataset:** The OSdataset [41] consists 2673 image pairs with a resolution of 512×512, and 10692 image pairs with the size of 256×256. This dataset collects images of scenes from cities around the world. The SAR images are captured by the sensor of Gaofen-3 (GF-3) multipolar C-band SAR satellite, and the optical images are obtained from the Google Earth platform with a spatial resolution of 1m. In experiments, 2673 image pairs are used for evaluation. The ratio of the training, validation and testing sets is 7:2:1. The

**Table 1: The comparison of the state-of-the-arts on SEN1-2 dataset.**

| Methods | CMR(T=5) | RMSE(T=5) | RMSE(All) | Time(s) |
|---------|----------|-----------|-----------|---------|
| NCC | 0.4068 | 2.2905 | 38.3845 | 60.1576 |
| NMI | 0.5739 | 1.3213 | 26.8528 | 86.2218 |
| CFOG | 0.6667 | 1.5395 | 19.1622 | 0.1672 |
| RIFT | 0.8043 | 1.6044 | 15.7632 | 0.2114 |
| Psiam | 0.6884 | 1.986 | 22.2338 | 95.4723 |
| VSMatch | 0.7174 | **1.3156** | 20.6323 | 87.155 |
| OSMNet | 0.9168 | 2.3086 | 4.7457 | 0.0568 |
| MARU-Net | 0.9056 | 1.4007 | 5.2601 | **0.0346** |
| Ours | **0.9302** | 1.3496 | **4.5227** | 0.0621 |

SAR images with the size of 512×512 are chosen as the search image, and the optical images cropped as 256×256 are treated as the template.

### 4.2 Experimental Settings

*Implementation details:* The method is implemented based on the PyTorch framework, and run on Nvidia Geforce RTX4090 GPU and Core i7-12700KF CPU. The feature stride of the autoencoder is 4. The feature strides of each feature layer in the denoising U-Net are {1, 2, 4, 8}. The gated dynamical sparse cross-attention module is used at the end of the feature layer with stride of {2, 4, 8}. The sparsity parameter $\gamma$ is fixed to 1 when feature stride is 8, since this layer features have rich semantics and few tokens. To train the denoising U-Net, we set the minimum time step $t$ to 0.001. The model is trained using the AdamW optimizer for 30k iterations. The decay rate ranges from 5e-5 to 5e-6.
*Evaluation metrics:* In experiments, the Root Mean Square Error (RMSE) and Correct Matching Rate (CMR) are used as evaluation metrics. The RMSE measures the average Euclidean distance between the prediction and the ground-truth. The CMR denotes the correct matching rate when the RMSE is less than a given threshold $T$, denoted as CMR(T). In the heterogenous reomote sensing image matching task, the matching error is less than or equal to 5 pixels is regarded as the successful matching. Therefore, we choose CMR (T=5) and RMSE (T=5) to evaluate the proposed approach. To further evaluate the overall matching performance, the average mean squared error of all samples, RMSE (all), is also adopted.

### 4.3 Comparison with State-of-the-arts

*Quantitative comparison.* To evaluate the performance of the proposed method, we compare it with state-of-the-arts on SEN1-2 and OSdataset. The compared approaches include traditional methods, NCC [20], NMI [2], CFOG [42], RIFT [21], and deep-learning-based methods, PSiam [19], VSMatch [9], SCMatch [26], OSMNet [44], MARU-Net [15].
Table 1 shows the matching performance of state-of-the-arts on SEN1-2 dataset. It is seen that the proposed method achieves 93.02% on CMR(T=5), performing best. Compared to the state-of-the-art approach OSMNet, our method improves 1.34%. The deep-learning based method, VSMatch, achieves the best RMSE(T=5) of 1.3156, which has a slight improvement of 0.034 compared to our

**Table 2: The comparison of the state-of-the-arts on OS-dataset.**

| Methods | CMR(T=5) | RMSE(T=5) | RMSE(All) | Time(s) |
|---|---|---|---|---|
| NCC | 0.1083 | 2.8915 | 79.7331 | 1034.1178 |
| NMI | 0.275 | 2.0746 | 62.1353 | 1250.2496 |
| CFOG | 0.5417 | 1.5922 | 22.9735 | 1.1465 |
| RIFT | 0.7583 | 1.9245 | 17.0226 | 6.6118 |
| Psiam | 0.5128 | 1.8952 | 27.6463 | 519.1755 |
| VSMatch | 0.6496 | 1.7437 | 24.2316 | 451.2569 |
| SCMNet | 0.7833 | **1.318** | 12.1056 | 63.4261 |
| OSMNet | 0.8043 | 2.4922 | 9.2731 | 0.0913 |
| MARU-Net | 0.8357 | 2.2495 | **6.9049** | **0.0895** |
| Ours | **0.8491** | 2.29 | 7.61 | 0.1093 |

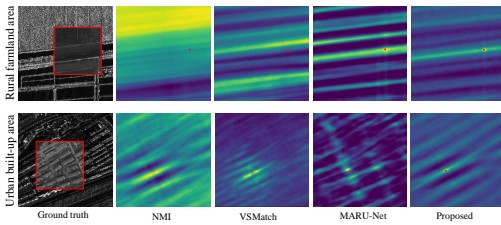

**Figure 6: The comparison of two samples' matching similarity map.**

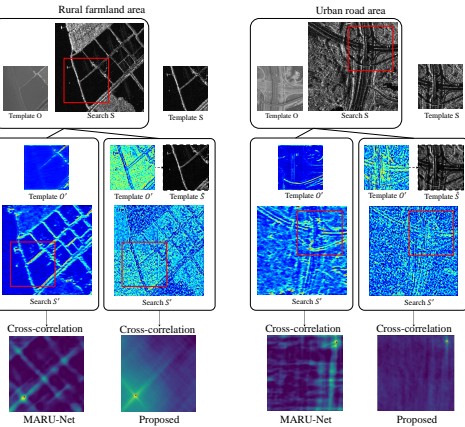

**Figure 7: The visualization comparison of extracted features and similarity maps.**

algorithm. For the matching time, the feature-based approaches, CFOG and RIFT, and the deep-learning based methods, OSMNet, MARU-Net, and our methods, takes less time. The MARU-Net only takes 0.0346s.

Compared to the overall performance on the SEN1-2 dataset, the OSdataset has a drop, since images in the OSdataset have higher spatial resolution, containing noises. Table 2 compares the matching results on the OSdataset. It is obvious that the region-based methods, NCC and NMI, perform poorly and take more time. The NCC only obtains 10.83% on CMR(T=5), and 2.8915 RMSE(T=5). Compared to region-based methods, the feature-based methods achieve a large improvement, where RIFT has 75.83% on CMR(T=5). The learning-based algorithms, Psiam, SCMNet and VSmatch are under 80% on the CMR(T=5). Whereas, the SCMNet achieves the best result on RMSE(T=5), reaching 1.318. Moreover, the three methods have quite long inference times compared to other learning-based and feature-based methods, since they slide the template by pixel over the search image and feed into the matching model to compute the similarity of matches. On the contrary, OSMnet, MARU-Net and our method take very short matching time. Furthermore, our method achieves 84.91% on CMR(T=5), having a 1.34% gains compare to the state-of-the art approach MARU-Net.

***Qualitative comparison of similarity maps.*** Figure 6 qualitatively shows the similarity maps produced by different methods on the two samples. In the similarity map, the higher the response value, the brighter the color. The location corresponding to the peak value is the best matching position. The red boxes in the first column 'Ground truth' denote the ground-truth matching areas. The red dots in other approaches represent the ground-truth offset coordinates in the matching map.

The first-row sample shows a pair of optical-SAR image in the rural farmland. The image has little texture and multiple similar regions, which makes optical-SAR matching be challenging. The comparison methods obtain similar responses in parallel linear directions, which is difficult to produce focused similarity maps. Especially, there exists a large highlight regions of similarity map obtained by NMI. VSMatch fails to correctly focus on, and the focused peak of MARU-Net is relatively unremarkable. Whereas the proposed method has low response values in non-matched regions and has a relatively prominent single peak.

The second-row sample depicts an urban area with dense geometrically similar buildings. The NMI and VSMatch are shifted from the correct matching point, where the VSMatch shows an undesired neighbouring bimodal characteristic. The peaks of the proposed method and MARU-Net correspond to the matched regions. But the response values of the similarity map achieved by our method are smoother in the non-matched region, which is due to the fact that generating the same modality features are more distinguishable.

***Qualitative comparison of matching features.*** Figure 7 visualizes the two samples' template feature maps (Template $O'$) and search feature maps (Search $S'$) obtained by the most representative state-of-the-art method MARU-Net and our approach. The first sample is a rural farmland area with less texture, and the second sample is a complex urban road area. Compare the feature maps and similarity maps produced by the two methods, it is observed that both the template and search feature maps of MAR-Net have very sparse texture features (especially for the first textureless regions) and inconsistent visualization, even though it attempts to map them into a shared feature space. These unreliable features lead to high correlation in non-matching regions. In contrast, the template and search feature maps generated by our method maintains all the texture information of the original image. This is due to the fact that our method focuses on generating feature maps of the same modality in latent space, which is more interpretable, and ensures that the similarity maps are single-peaked.

**Table 3: The ablation study on the SEN1-2 dataset.**

| Base | CRB | ACB | AG | CMR(T=5) | RMSE(T=5) | Time(s) |
|------|-----|-----|-----|----------|-----------|---------|
| √ | | | | 0.8457 | 2.1341 | 0.1021 |
| | √ | | | 0.8672 | 1.8904 | **0.0542** |
| | √ | √ | | 0.9123 | 1.5301 | 0.0567 |
| | √ | | √ | 0.8931 | 1.6577 | 0.0605 |
| | √ | √ | √ | **0.9302** | **1.3496** | 0.0621 |

**Table 4: The effect of the sparsity parameter $\gamma$ on the SEN1-2. GDSCs is added to features with strides of {2, 4, 8}.**

| $\gamma$ | CMR(T=5) | RMSE(T=5) | RMSE(All) | Time(s) |
|----------|----------|-----------|-----------|---------|
| $\gamma$=(1, 1, 1/2) | 0.9175 | 1.4021 | 4.8412 | 0.0937 |
| $\gamma$=(1, 3/4, 3/8) | **0.9302** | **1.3496** | **4.5227** | 0.0621 |
| $\gamma$=(1, 1/2, 1/4) | 0.8843 | 1.6545 | 5.0463 | 0.0441 |
| $\gamma$=(1, 1/4, 1/8) | 0.8254 | 1.9874 | 6.7734 | **0.0312** |

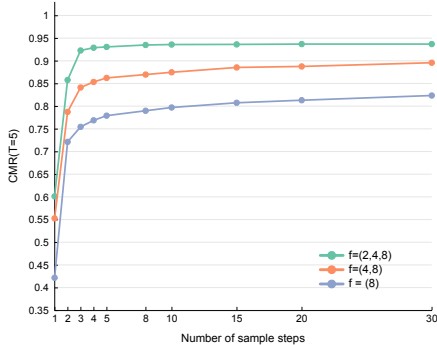

**Figure 8: The effect of the number of GDSCs and sample steps on CMR(T=5) with the SEN1-2.**

## 4.4 Ablation Experiments

To verify the effect of each proposed components and the parameters on the matching performance, we conduct ablation experiments on the SEN1-2 dataset.

***The effect of key components.*** Based on the LDM, we proposed three key components to improve the cross-modal remote sensing image matching performances. To verify the effectiveness of each proposed component, we conduct ablation experiments on the coarse-grained routing block (CRB), the attention calibration block (ACB), and the attention gate (AG). For the baseline and our approach, the denoising sampling step $t$ is set to 5, and the cross-attention module is added to feature layers with strides of {2, 4, 8}. For our method, the sparsity parameter, $\gamma$, is set to (1, 3/4, 3/8). As shown in Table 3, the baseline only achieves 84.57% on the CMR(T=5). Compared to the cross-attention in the baseline, the proposed CRB outperforms the base attention by 2.15% on the CMR(T=5), reaching 86.72%. The attention calibration module obtains a remarkable gains of 4.51% on CMR(T=5), reaching 91.23%. The attention gate module improves 2.59% on CMR(T=5), achieving 89.31%. The proposed three modules achieve remarkable improvements on the CMR(T=5) and the RMSE(T=5). Although introducing three modules, the inference time is not significantly increased, since the proposed modules are lightweight.

***The effect of the number of GDSCs and sample steps.*** We investigate the effect of the number of GDSCs on the matching accuracy. In the denoising U-Net, the gated dynamical sparse cross-attention (GDSC) module are respectively introduced into the three feature layers corresponding to feature strides of {2, 4, 8}. The sparsity parameters $\gamma$ in the three feature layers are set as 1, 3/4, and 3/8, respectively. Figure 8 illustrates the effect of the number of the GDSC module. It is found that the CMR(T=5) improves with the increase of the number of GDSCs, which is due to the fact that a larger number of attention layers enable the network to capture fine-grained information.

In addition, it is observed that the denoising sample steps influence the matching performance. The higher the number of sample steps, the higher the value of CMR(T=5). Nevertheless, there are a few increases when the sample steps greater than 5. For instance, given the added feature layers of GDSC as feature strides of {2, 4, 8}, the CMR (T=5) only increases by 0.5% when the sampling steps increase from 5 to 10 (the green line in Figure 8), when the GDSC is introduced into features layers . To balance the trade-off between the speed and the performance, the sample steps is set as 5 in experiments.

***The effect of the sparsity parameter $\gamma$.*** We further analyze the effect of the sparsity parameter $\gamma$ on the matching accuracy and time. As shown in Table 4, the model performs best when the sparsity parameters $\gamma$=(1, 3/4, 3/8), where the CMR reaches to 93.02% and the RMSE is 1.3496. Compared to the best performance, the CMR(T=5) slightly decreases by 1.27% and the RMSE(T=5) increases by 0.0525 when $\gamma$=(1, 1, 1/2), since larger $\gamma$ tends to introduce more error messages and greater matching delays. The CMR(T=5) decreases by 4.59% when the sparse parameters $\gamma$=(1, 1/2, 1/4), and even decreased by 10.49% when $\gamma$=(1, 1/4, 1/8). This is due to the fact that too smaller $\gamma$ causes insufficient cross-modal information interaction, which make the model be difficult to converge and thus degrading the performance.

## 5 CONCLUSION

The paper proposes a dynamically conditioned diffusion model to achieve the interpretable and robust optical-SAR cross-modal image matching. Specifically, the gated dynamic sparse cross-attention module is used to guide the diffusion model to capture information from conditions through the efficient long-range cross-modal interactions, and thus filtering out outlier matching regions. In addition, the spatial position consistency constraint promotes the cross-attention features to perceive the spatial corresponding relation in different modalities, and improves the matching accuracy. Experimental results on two datasets show that the proposed method outperforms state-of-the-art approaches in terms of the matching accuracy and the interpretability. The study provides an exploration for future researches on the cross-modal image matching or registration under the diffusion models.

 

## Acknowledgments

This work is supported by the project of Science and Technology Development Plan in Hangzhou under Grant No. 202202B38, in part by the Fundamental Research Funds for the Central Universities under Grant No.XJSJ24071, in part by the Key Laboratory of Cognitive Radio and Information Processing, Ministry of Education under Grant No.CRKL230204, in part by the Fundamental Research Funds for the Central Universities under grant No.XJSJ24072, in part by the National Natural Science Foundation of China under grant No. 62302355.

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
