# OpenReview forum: "Interpretable Matching of Optical-SAR Image via Dynamically Conditioned Diffusion Models"
_acmmm.org/ACMMM/2024/Conference — MM2024 Poster_

### Official Review · Reviewer_t3a4 · 2024-05-01

**Rating:** 4
**Confidence:** 3

**Summary:**

The paper introduces a novel method for matching optical and Synthetic Aperture Radar (SAR) images using a Dynamically Conditioned Diffusion Model (DCDM). This approach is designed to address the significant challenge of matching these images due to their different imaging characteristics. The method utilizes a gated dynamic sparse cross-attention module to facilitate efficient interaction of features across modalities and a spatial position consistency constraint to enhance matching accuracy. Experimental results on two datasets show that the proposed method outperforms state-of-the-art approaches in terms of the matching accuracy and the interpretability.

**Strengths:**

1. Novel approach: The paper proposes a unique approach using diffusion models for cross-modal image matching, which is a relatively unexplored area in image processing.

2. Experimental validation: The proposed approach is evaluated on two challenging datasets, SEN1-2 and OSdataset, and demonstrates superior performance compared to existing techniques in terms of Correct Matching Rate.

3. Paper writing: The paper provides a detailed explanation of the construction of different modules, making it easy for researchers in the field to understand the work.

**Limitations:**

1. It appears that the method does not perform as well on the RMSE metric when compared to the best-performing methods in Table 2,  and could you provide a brief analysis of the reasons?

2. About the stated criterion ‘In the task of matching heterogeneous remote sensing images, a matching error of less than or equal to 5 pixels is considered a successful match’ In Evaluation metrics section. What is the basis for this matching error criterion? Can you provide relevant literature support or examples of the application of similar criteria in previous studies?

3. Lack of generalization analysis: The paper does not provide a generalization analysis of the proposed approach on other datasets, which would provide insights into the adaptability of the network to different scenarios.

4. The integration of various strategies like Dynamically Conditioned Diffusion Models, gated dynamic sparse cross-attention, and spatial position consistency constraints, while innovative, introduces considerable complexity. This blend of multiple advanced components may complicate model interpretation, implementation, and optimization. The lack of clarity in how these components interact could lead to potential inefficiencies or diminished performance gains due to overlapping features.

**Suitability:**

3

---

### Official Review · Reviewer_qe73 · 2024-05-19

**Rating:** 4
**Confidence:** 3

**Summary:**

In this manuscript, a dynamic conditional diffusion model is proposed, based on real interpretable and robust optical-SAR cross-modal image matching, mainly gated dynamic sparse cross-attention module and spatial position consistency constraint, and experiments on two data sets are carried out to prove the effectiveness of the proposed method.

**Strengths:**

1. The manuscriptdescribes the proposed method clearly, and the language is fluent and professional.
2.  The method is novel, based on diffusion model design, and cross attention is introduced into the proposed method to promote the interaction of multi-granularity features between cross-modal data.
3. Considering that the plain area in the remote sensing image is texture-free, Attention Gate is introduced to control the inflow of conditional information.
4.  In the experimental part, there are many methods of comparison, and explore the influence of super-parameters on the model.

**Limitations:**

1. Although a large number of methods are compared in quantitative experiments,
only one method has been used in the past two years, and the other methods are relatively old. In addition, there are few methods of visual comparison.
2. From tables 1 and 2, there is little difference in performance between the proposed method and the most advanced method, and it does not seem to be very convincing.
3. In Table 3, why does it take less time overall when modules are added to Base?
4.  In the The effect of key components of Section 4.4, the manuscript only describes the experimental results and lacks the explanation of the mechanism of the components.

**Suitability:**

2

---

### Official Review · Reviewer_nfJz · 2024-05-19

**Rating:** 2
**Confidence:** 4

**Summary:**

This paper introduces a dynamically conditioned diffusion model for optical-SAR image matching. Their approach presents a gated dynamic sparse cross-attention module to inject reliable conditional information and a spatial position consistency constraint to enhance the perception of matching positions. Experimental results demonstrate the effectiveness of the proposed approach on two datasets.

**Strengths:**

1. The writing is clear with clear figures.
2. The proposed two modules are demonstrated concisely and easy to follow.

**Limitations:**

1. While the testing time is provided for comparison, the paper lacks discussion on memory usage and network parameters of the proposed method, which should be included for comprehensive analysis.
2. The paper introduces an interpretable and robust matching method, yet it fails to provide theoretical or experimental justification for its interpretability and robustness.
3. The comparison in the paper only includes one IEEE JSTRAR paper from 2023. It would be beneficial to include comparisons with recent state-of-the-art (SOTA) methods published in top journals or conferences in 2023 or 2024, if available.
4. Tables 1 and 2 indicate that the proposed method does not achieve state-of-the-art experimental results.

**Suitability:**

1

---

### Official Review · Reviewer_Q5Zg · 2024-05-24

**Rating:** 4
**Confidence:** 4

**Summary:**

This paper formulates the optical-SAR cross modal image matching as a dynamically conditioned diffusion model (DCDM) to learn the posterior distribution of regions with dense correspondences. Firstly, the Gaussian noise is continuously added to the ground-truth SAR template in each diffusion step, and the SAR template is generated by training a U-Net-based denoising network. To generate more realistic SAR images, the corresponding optical template is adopted as a condition to provide the scene content. Additionally, the search SAR image is introduced to provide the texture details for the accurate matching. The gated dynamic sparse cross-attention module and the spatial position consistency constraint are applied to achieve the effective and efficient cross-modal feature interaction and aggregation. Finally, FFT-based NCC is adopted to perform matching between the generated SAR template and the search SAR in the latent space. Experimental results also show that the proposed method outperforms state-of-the-art methods.

**Strengths:**

1. The introduction of the gated dynamic sparse cross-attention module enhances the efficiency and effectiveness of feature interaction between cross-modal data. This module helps filter out outlier matching regions, which is crucial for accurate matching.
2. The paper introduces a spatial position consistency constraint that promotes better perception of spatial correspondences across different modalities, thereby improving the matching precision.
3. This paper proposes an end-to-end cross-modal image matching framework, which not only translates cross-modal images, but also completes the pixel-level matching in the latent space.

**Limitations:**

1. While the paper claims that the proposed method reduces computational complexity by operating in the latent space, the overall framework, including the diffusion process and the gated dynamic sparse cross-attention, may still be computationally intensive, especially for large datasets. I hope the authors can provide an analysis of the model complexity and the number of parameters.
2. The authors should carefully review the entire manuscript to improve language accuracy and avoid grammatical errors such as "The study provide" and similar issues.
3. For Figure 1, the authors should make the "Divide" and other labels clearer. Additionally, it would be best to provide some relevant legends in Figure 3.

**Suitability:**

3

---

### Meta-Review · Area_Chair_Z82e · 2024-07-02

**Recommendation:** Accept (Poster)
**Confidence:** 5

**Metareview:**

In this paper, the authors propose a dynamic conditional diffusion model for optical-SAR image matching. It introduces a gated dynamic sparse cross-attention module to inject reliable conditional information and a spatial position consistency constraint to enhance the perception of matching positions. The experimental results prove the effectiveness of the proposed method on two datasets. The paper is overall well written, but some grammatical errors need to be corrected. Its novelty is good. The authors are expected to analyze its generalization as well as the model complexity, and add more comparison results with recent SOTA works published in 2023 and 2024.

After rebuttal, this paper receives 3 positive reviews and 1 negative review, so I recommend the acceptance of this paper.